# *Vibrio cholerae* biofilms use modular adhesins with glycan-targeting and nonspecific surface binding domains for colonization

Xin Huang [1,2,11], Thomas Nero[1,11], Ranjuna Weerasekera [3], Katherine H. Matej [1], Alex Hinbest[3], Zhaowei Jiang [1], Rebecca F. Lee[4], Longjun Wu[5,10], Cecilia Chak[1], Japinder Nijjer[1], Isabella Gibaldi[3], Hang Yang[3], Nathan Gamble [3], Wai-Leung Ng[6], Stacy A. Malaker [2], Kaelyn Sumigray[4,7,8], Rich Olson [3] ✉ & Jing Yan [1,9] ✉

Bacterial biofilms are formed on environmental surfaces and host tissues, and facilitate host colonization and antibiotic resistance by human pathogens. Bacteria often express multiple adhesive proteins (adhesins), but it is often unclear whether adhesins have specialized or redundant roles. Here, we show how the model biofilm-forming organism *Vibrio cholerae* uses two adhesins with overlapping but distinct functions to achieve robust adhesion to diverse surfaces. Both biofilm-specific adhesins Bap1 and RbmC function as a "double-sided tape": they share a β-propeller domain that binds to the biofilm matrix exopolysaccharide, but have distinct environment-facing domains. Bap1 adheres to lipids and abiotic surfaces, while RbmC mainly mediates binding to host surfaces. Furthermore, both adhesins contribute to adhesion in an enteroid monolayer colonization model. We expect that similar modular domains may be utilized by other pathogens, and this line of research can potentially lead to new biofilm-removal strategies and biofilm-inspired adhesives.

Bacterial biofilms are surface-attached communities of bacterial cells enclosed in an extracellular matrix[1]. Biofilms represent an important lifestyle niche for bacteria in the environment as well as a serious threat to human health due to their role in persistent infections and contamination of medical devices[2–4]. One key evolutionary advantage provided by biofilm formation is surface adhesion, whereby bacteria physically associate with essential environmental and host surfaces to prioritize nutrient uptake and to resist environmental stressors[5,6]. Among the components of the biofilm matrix, exopolysaccharides and accessory proteins have been suggested to play predominant roles in biofilm adhesion to both biotic and abiotic surfaces[7]. However, how they function at the molecular level remains largely unknown. One

[1]Department of Molecular, Cellular and Developmental Biology, Yale University, New Haven, CT, USA. [2]Department of Chemistry, Yale University, New Haven, CT, USA. [3]Department of Molecular Biology and Biochemistry, Molecular Biophysics Program, Wesleyan University, Middletown, CT, USA. [4]Department of Genetics, Yale School of Medicine, New Haven, CT, USA. [5]Department of Ecology and Evolutionary Biology, Yale University, New Haven, CT, USA. [6]Department of Molecular Biology and Microbiology, Tufts University School of Medicine, Boston, MA, USA. [7]Yale Stem Cell Center, Yale School of Medicine, New Haven, CT, USA. [8]Yale Cancer Center, Yale School of Medicine, New Haven, CT, USA. [9]Quantitative Biology Institute, Yale University, New Haven, CT, USA. [10]Present address: Department of Ocean Science and Hong Kong Branch of the Southern Marine Science and Engineering Guangdong Laboratory (Guangzhou), The Hong Kong University of Science and Technology, Hong Kong SAR, Guangzhou, Hong Kong SAR. [11]These authors contributed equally: Xin Huang, Thomas Nero. ✉e-mail: rolson@wesleyan.edu; jing.yan@yale.edu

major puzzle concerns what biochemistry makes these adhesins specific to the biofilm lifestyle: efforts in the field have focused on the initial stage, during which pili, flagella, and other adhesins contribute to the attachment of *individual* cells[8]. In contrast, little is known about how matrix-encased cells *collectively* adhere to surfaces in mature biofilms and to what extent this mechanism differs from the single-cell case. Such a mechanistic understanding is critically relevant for designing new biofilm removal strategies that target biofilm-surface interactions as an alternative to antibiotic treatments and for creating new biofilm-inspired materials[9].

Here, we address these questions using *Vibrio cholerae*, the causal agent of pandemic cholera and a model biofilm-forming organism[10,11]. Biofilm formation has been shown to be important for *V. cholerae* adhesion to chitinous surfaces in the aquatic environment;[11] although not absolutely essential for colonizing the mammalian intestine, biofilm formation provides advantages for *V. cholerae* to thrive in this niche[12–14]. Even though surface attachment can be achieved by several other adhesive factors that function at the individual cell level[15–18], we focus here on the molecular mechanism of *biofilm-specific* surface adhesins. The major biofilm phenotype in *V. cholerae* is dependent on Vibrio polysaccharide (VPS)[19], which possesses a unique tetrasaccharide repeating unit[20] and plays the primary role in controlling the structural integrity of *V. cholerae* biofilms[21].

In addition to VPS, *V. cholerae* possesses two putative surface adhesins that function specifically in the biofilm context, Bap1 and RbmC[22], which were thought to be largely redundant. The double-deletion mutant shows significantly impaired colony rugosity, an inability to adhere to abiotic surfaces, and a colonization defect in *Drosophila melanogaster*[22–25]. Puzzlingly, immunolabeling results show that their spatial distribution within biofilms differs significantly: while the Bap1 signal is concentrated at the biofilm-substrate interface, RbmC forms envelope-like structures surrounding the biofilm together with VPS and Bap1[26]. It is unclear if and how the two adhesins function differently and why they are biofilm specific as opposed to classical adhesins that function at a cellular level. To pinpoint the structural basis underlying the adhesion mechanism, we recently obtained crystal structures of Bap1 and a lectin domain of RbmC[27,28]. Here, we combine insights from prior structural work with functional assays to show how *V. cholerae* uses two biofilm-specific adhesins with overlapping but distinct functions to achieve robust, diverse surface adhesion.

## Results

### Bap1 and RbmC contain modular and overlapping domains

Figure 1a illustrates the domain organization of the two biofilm adhesins based on the crystal structure of Bap1 (Fig. 1b) and homology

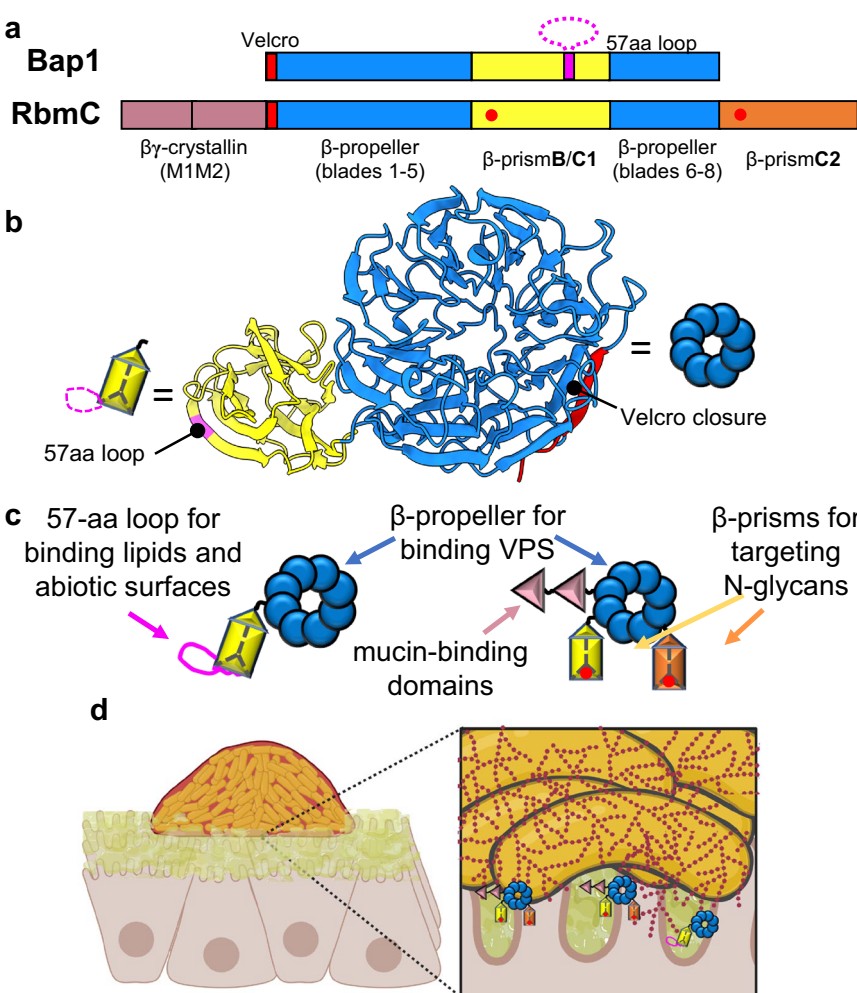

**Fig. 1 | Bap1 and RbmC contain modular and overlapping domains for surface adhesion. a** Schematic of Bap1 and RbmC structural domains. Red dots correspond to the positions of the N-glycan binding pockets. **b** Crystal structure of Bap1 without the 57aa loop and the corresponding cartoon for each domain. The position of the 57aa loop (magenta) and the β-strand from β-propeller blade 1 that contributes to the Velcro closure (red) are indicated. **c** Schematic and hypothetical functions of domains in Bap1 (*Left*) and RbmC (*Right*). **d** Working model. Both proteins function as "double-sided tape" for *V. cholerae* biofilms: they share a conserved β-propeller that binds to VPS. Bap1 adheres to lipids and abiotic surfaces primarily via the 57aa loop while RbmC mainly mediates binding to host surfaces via N-glycan-binding pockets in its β-prism**C**s and the mucin-binding domains M1M2. Created with Biorender.com.

modeling of RbmC. Both proteins contain a conserved β-propeller domain with eight blades (~70% sequence identity between RbmC and Bap1 β-propellers) containing a Velcro closure β-strand that holds the first and last blades together forming a ring. Both proteins incorporate at least one β-prism domain, a putative sugar-binding fold widely found in nature and particularly in plant lectins. Interestingly, Bap1 contains a single β-prism domain while RbmC contains two, and the sequence of Bap1's β-prism (abbreviated as β-prism**B**) diverges from those of the two β-prisms in RbmC (abbreviated as β-prism**C**s)[27]. Most notably, β-prism**B** contains an additional 57-amino acid (aa) sequence whose structure and function are unknown; this 57aa loop needed to be removed to produce soluble Bap1 protein for structure determination[28]. RbmC additionally has two tandem N-terminal β/γ-crystallin domains. The relative and collective roles of the various domains of Bap1 and RbmC remain unknown. To test the differing contributions of the two adhesins and the function of their constituent domains, we generated *V. cholerae* mutants in which one or multiple domains of Bap1 and RbmC are deleted or modified in the *native* locus (Supplementary Fig. 1), and whenever possible, we purified the corresponding mutant proteins or domain(s) from *E. coli* attached to a GFP_UV label[29]. To focus on the biochemical mechanisms of biofilm adhesion rather than the effects of gene regulation, we mainly use a strain locked in a high cyclic diguanylate level that constitutively produces biofilms[30].

## A conserved β-propeller domain anchors Bap1 and RbmC to VPS

We first asked the question of what makes Bap1 and RbmC biofilm-specific and hypothesized that the key lies in the connection between the adhesins and the main structural component of the *V. cholerae* biofilm, VPS, through the conserved β-propeller domain. To support this hypothesis, we used several complementary methods. First, we added *E. coli*-purified domain(s) from Bap1 to mature *V. cholerae* biofilms and observed that any construct containing an intact β-propeller

domain shows a positive staining signal; this staining disappears upon deletion of *vpsL*, one of the key biogenesis genes necessary for VPS production (Fig. 2a). Second, we tagged Bap1 with a 3×FLAG tag at its C-terminus and performed immunostaining; consistent with the prior literature, we observed envelope-like structures surrounding the biofilm cluster (Fig. 2b) and notably, this staining pattern only requires the presence of the β-propeller domain (Supplementary Fig. 2a). These results confirm that the β-propeller binds to VPS, either directly or indirectly, but not to the *V. cholerae* cell surface, thus explaining why Bap1 and RbmC are involved in biofilm adhesion but not the initial attachment of cells to substrates[11].

To further investigate this interaction, we attempted to generate a Bap1 mutant in *V. cholerae* without the β-propeller domain; unfortunately, this construct is not properly secreted (Supplementary Fig. 2b), so we pursued an alternative strategy. We reasoned that deleting the Velcro closure should result in a secretable Bap1 with a nonfunctional β-propeller domain interrupting VPS interactions. Indeed, this Bap1_ΔVelcro mutant is successfully secreted, but defective in adhering *V. cholerae* biofilms to the surface (Supplementary Fig. 2c–f). Interestingly, immunostaining shows that the Bap1_ΔVelcro mutant still properly localizes at the biofilm-glass interface, but in the bulk of the biofilm it forms puncta-like structures on the order of 250–550 nanometers (Fig. 2b, Supplementary Fig. 2g, h), in contrast to the envelope pattern formed by wild-type (WT) Bap1[26]. These results confirm that the VPS-β-propeller interaction is important for proper functioning and spatial localization of Bap1.

To test if the binding between VPS and the β-propeller domain is direct or through other intermediate factors, we performed an electrophoretic mobility shift assay (EMSA) with purified VPS and various purified domains of Bap1 with a GFP-tag under non-denaturing conditions (Fig. 2c). For constructs that contain the β-propeller, we observe a decrease in the intensity of the unbound protein bands and simultaneously, the emergence of slower moving, high molecular

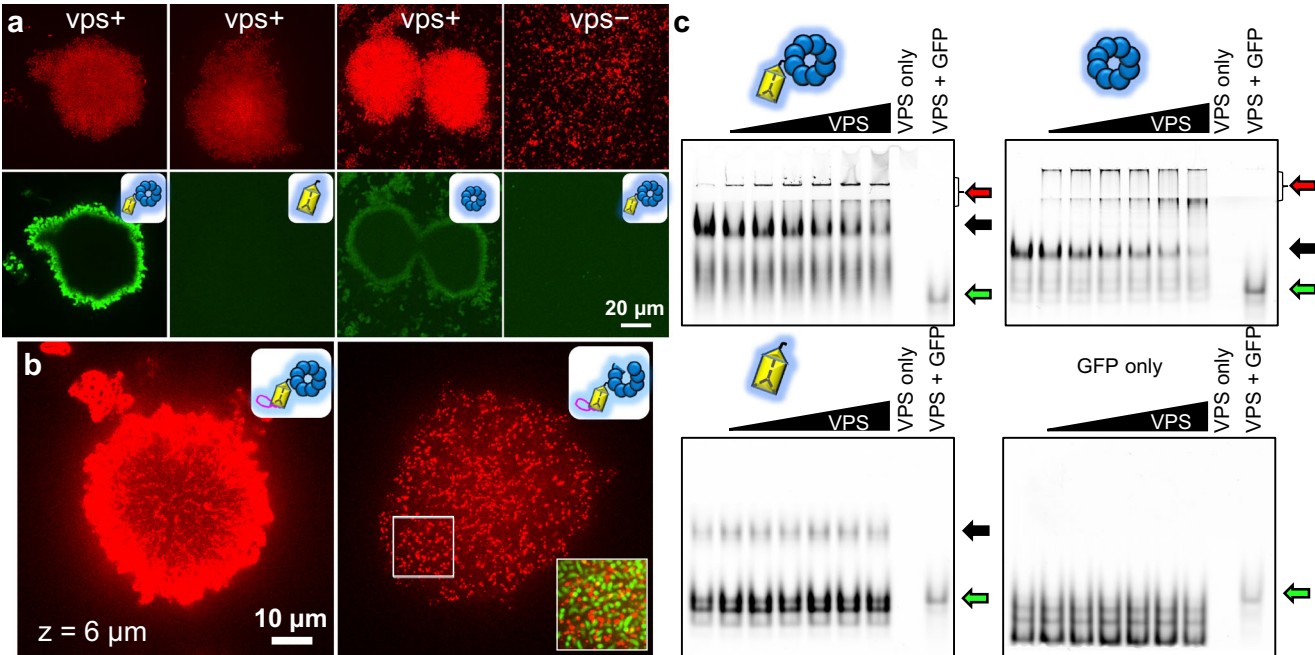

**Fig. 2 | A conserved β-propeller domain anchors Bap1 and RbmC to VPS. a** *V. cholerae* biofilms expressing mScarlet-I (*Top*) incubated with 1 μM purified and GFP-tagged proteins (*Bottom*) with the indicated domain(s). The VPS⁻ control was performed with Δ*vpsL* cells unable to produce VPS. **b** Cross-sectional images (at z = 6 μm) of biofilms from cells expressing wild-type Bap1 (*Left*) and Bap1_ΔVelcro (*Right*), both tagged with a 3×FLAG tag at the C-terminus and stained with an anti-FLAG antibody conjugated to Cy3. Inset: magnified image of the region highlighted

by the white box, with the cell signal (mNeonGreen) overlaid with Cy3. **c** EMSA images showing the binding of purified Bap1's β-propeller to purified VPS. Red arrows = protein-VPS complex, black arrows = unbound proteins, green arrows = free GFP. See Supplementary Fig. 2i for positive control and results from RbmC's β-propeller. When present, the protein amount in a lane is 5 μg. The VPS amount is 0, 0.0625, 0.125, 0.25, 0.50, 1, 5, 5, 5 μg in each lane from left to right.

weight bands that do not enter the gel, indicating the formation of large aggregates[31]. In contrast, the β-prism**B** and GFP-only negative controls do not show this behavior. We confirmed this same observation with RbmC's β-propeller (Supplementary Fig. 2i). These results provide evidence of direct binding between VPS and the β-propeller, which underlies why the two adhesins are specific to the biofilm lifestyle.

## Bap1 possesses a unique 57aa loop with broad, nonspecific adhesion

We next probed the function of other domains in the two adhesins. To quantitatively assess the adhesive ability of Bap1 and RbmC mutants, we first performed a standard crystal violet assay for single and double deletions of *bap1* and/or *rbmC*, and consistent with literature findings[23,24], we observed a predominant contribution of Bap1 to adhesion to abiotic surfaces such as glass and plastics (Supplementary Fig. 3a, b). Therefore, we first focused on the contribution of Bap1's different structural domains to surface adhesion in a Δ*rbmC* background. We grew biofilms of different mutants overnight on submerged glass surfaces and measured the fraction of adhered biomass remaining after vigorous washing for each mutant strain using confocal microscopy (Fig. 3a). To amplify differences in adhesion strength between the mutants, we challenged strains with increasing concentrations of Bovine Serum Albumin (BSA), as is commonly done to increase the stringency of washing in biochemical assays. We observe that the mutant missing the 57aa loop ($bap1_{\Delta 57aa}$) is unable to remain adhered in the presence of BSA (open circles, Fig. 3a), consistent with its morphological differences compared to biofilms with a WT Bap1 (Supplementary Fig. 3c). On the other hand, a Bap1 construct in which we deleted the β-prism**B** and directly attached the 57aa loop to the β-propeller ($bap1_{\Delta prismB+57aa}$) is fully functional (closed circles, Fig. 3a). These results indicate that the 57aa loop, absent in RbmC, is the main contributor of biofilm adherence to abiotic surfaces in *V. cholerae*. Removal of both the β-prism**B** and the 57aa loop results in minimal adhesion ($bap1_{\Delta prismB}$, open squares, Fig. 3a). This behavior is reproducible on other abiotic surfaces such as polystyrene and modified glasses (Supplementary Fig. 3d). The loss of function in the defective mutants is unlikely due to changes in the production or secretion level of the mutant protein (Supplementary Fig. 4a–d), and in all defective strains the adhesion defects can be rescued by WT Bap1/RbmC expressed from a plasmid (Supplementary Fig. 4e).

To probe the underlying mechanism of these adhesion defects, we asked if our mutant constructs exhibited an altered localization pattern (Fig. 3b, c). Consistent with previous work, WT Bap1 is particularly concentrated at the biofilm-substrate interface while also forming envelope structures along with VPS around cell clusters[23,24,26]. In contrast, the $Bap1_{\Delta 57aa}$ mutant abolishes the immunosignal at the biofilm-glass interface while retaining staining at the periphery of the biofilm, suggesting that the mutant protein has a diminished tendency to adsorb to glass surfaces but has not lost its VPS-binding. The defect in adsorption likely underlies why $Bap1_{\Delta 57aa}$ is defective in anchoring biofilms to a substrate. Additionally, the $Bap1_{\Delta prismB+57aa}$ mutant still exhibits a strong signal at the biofilm-substrate interface, consistent with its full function in adhesion assays. These observations reinforce the idea that the 57aa loop in Bap1 is the primary player in promoting *V. cholerae* biofilm adhesion to abiotic surfaces.

A closer look at the sequence of the 57aa loop reveals an abundance of aromatic and positively-charged residues (Fig. 3d) such as tyrosine (8.8%), tryptophan (8.8%), and lysine (12.7%). Both the function and the sequence of the 57aa loop is reminiscent of bivalve adhesion proteins: extensive work on adhesive mussel foot proteins (Mfps) highlights the importance of positively-charged residues, in conjunction with adjacent aromatic residues, in promoting adhesion to abiotic surfaces in aquatic environments[32,33]. The cap of the β-prism**B** domain adjacent to the 57aa loop also shares similar features,

although to a lesser extent[28]. Therefore, we suggest that the β-prism**B**'s cap together with the 57aa loop may form a continuous, positively-charged and aromatic surface enabling Bap1 to adhere nonspecifically to environmental surfaces in a manner similar to Mfps.

To further demonstrate the adhesive properties of the 57aa loop, we chemically synthesized the 57aa peptide N-terminal labeled with fluorescein isothiocyanate (FITC) and developed a protocol to visualize and quantify its physical adsorption to microbeads using fluorescence (Fig. 3e, f, Supplementary Fig. 3e, f). Compared to the FITC control, we observed a strong tendency for the 57aa peptide to spontaneously coat silica beads. We repeated the adsorption assay with lipid-coated beads and observed an even stronger adhesion signal, suggesting that the 57aa loop may also allow *V. cholerae* to adhere to the plasma membrane of epithelial cell surfaces during infection. Indeed, the FITC-labeled 57aa stains the entire cell surface in human intestinal epithelial slices (Fig. 3g).

## RbmC targets host cell surfaces via N-glycan and mucin-binding domains

Differences in the sequences of β-prism**B** and **C**s lead us to hypothesize that RbmC and Bap1 will behave differently when interfacing with host surfaces that *V. cholerae* might encounter during infection. Our previous glycan array analysis indicated specific binding of β-prism**C**s to complex N-glycans prevalent on the surface proteins of host cells[27]. Subsequently, we showed that β-prism**C**s can bind to the core branch-region of N-glycans with nanomolar affinity, and we obtained crystal structures of N-glycan fragments bound to β-prism**C2**[27]. Here, we first show that purified, GFP-tagged β-prism**C1** and **C2**, but not β-prism**B**, shows specific binding to Caco-2 cells (Fig. 4a, Supplementary Fig. 5a). This binding is a direct consequence of N-glycan targeting because mutation of the key aspartate residue in the N-glycan binding pocket (for example D853 in β-prism**C2**) abolished Caco-2 cell binding (Fig. 4a). Consistent with the Caco-2 binding results, we also observed binding of the β-prism**C**s, but not β-prism**B**, to the surface of human small intestine jejunum cells (Supplementary Fig. 5b).

To test the glycan-binding ability of *V. cholerae* biofilms in vitro, we developed an assay by coating glass substrates with asialofetuin, an N-glycosylated protein[34]. Interestingly, the surface staining pattern of Bap1 and RbmC is reversed in the presence of asialofetuin compared to that on bare glass (Fig. 4b, c, Supplementary Fig. 6a): RbmC shows a strong signal concentrated at the surface of the asialofetuin-coated substrate, whereas Bap1 only shows a peripheral staining pattern around cell clusters, consistent with the different targets of these two proteins. The surface-concentrated RbmC signal on asialofetuin can be abolished by disrupting the N-glycan binding pocket (Supplementary Fig. 6b, c).

Besides differences in the β-prism domains outlined above, RbmC additionally contains two tandem β/γ crystallin domains (denoted as M1M2) at its N-terminus with homology to the C-terminal domain of StcE, a mucinase from *E. coli* (67.7% identity between M1 and M2, 58.5% identity between M1/M2 and StcE C-terminal domain)[35,36]. By staining human intestinal slices with fluorescently labeled proteins[37], we find that both RbmC$_{M1M2}$ and StcE$_{C-term}$ label the mucus layer surrounding the epithelial surface and especially Goblet cells that secrete mucus (Fig. 4d). Furthermore, the M1M2 signal largely colocalizes with that from a MUC2 antibody (Fig. 4e). A gel-shift assay similarly shows that both M1M2 and StcE$_{C-term}$ bind to commercially available bovine submaxillary mucin (Supplementary Fig. 5c). The mucus-targeting ability demonstrates another mechanism by which RbmC may contribute to biofilm adhesin during host colonization.

## Bap1 and RbmC contribute to colonization in an enteroid monolayer model

To evaluate the validity of our findings in vivo, we employed a recently developed model of a two-dimensional intestinal epithelial (enteroid)

monolayer[38]. This monolayer was derived from mouse jejunum crypts (Fig. 5a) and contains gut-like differentiated cell types including mucus-secreting goblet cells, crypt-like domains, and villus-like regions (Supplementary Fig. 7), therefore presenting biochemical

features of the gut environment likely encountered by *V. cholerae*. The exposed apical surface allows us to study the encounter between *V. cholerae* and the enteroid monolayer with high-resolution imaging (Fig. 5b). With this setup, we found that the presence of the two

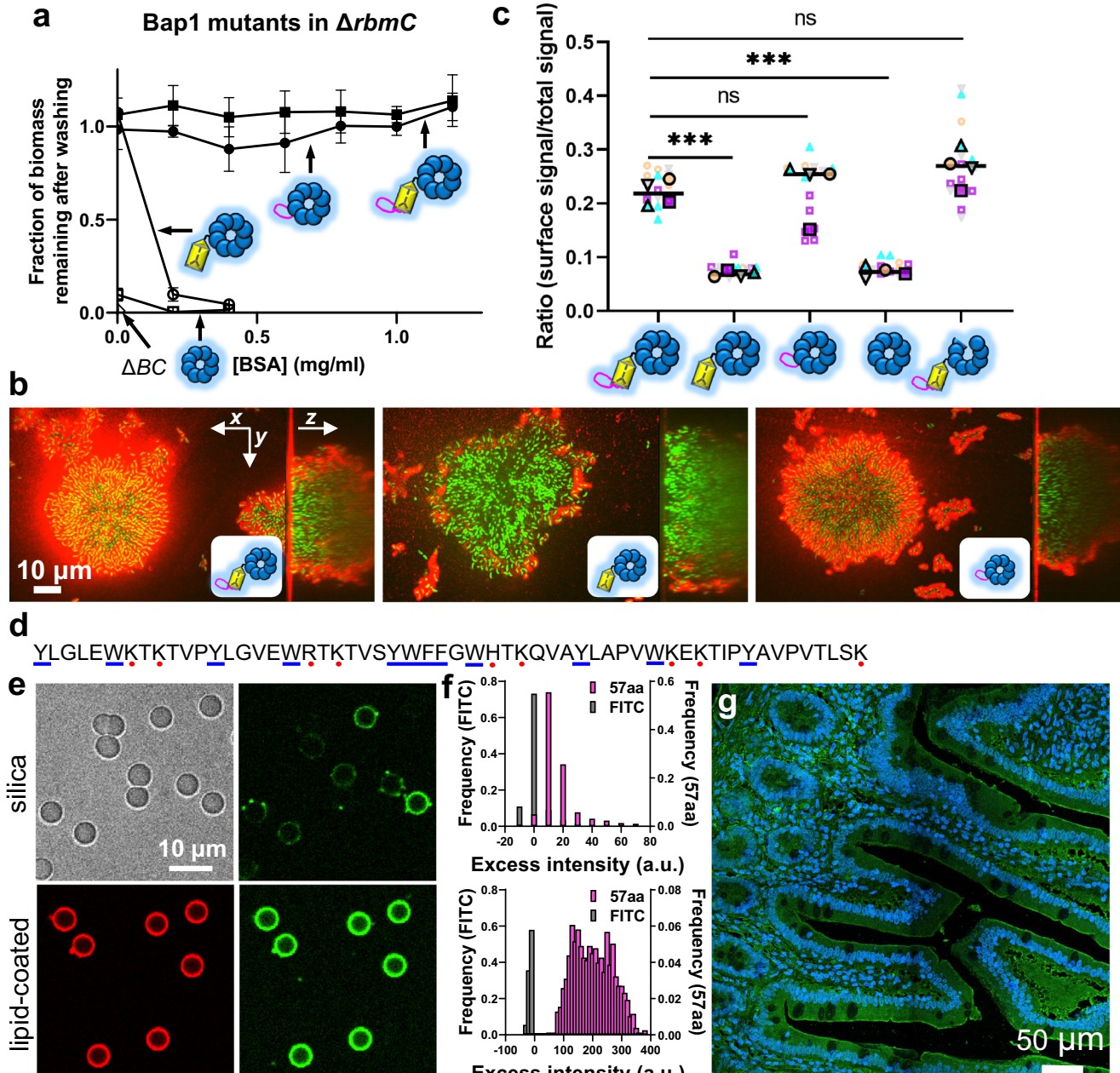

**Fig. 3 | Bap1 possesses a unique 57-amino acid loop with broad, nonspecific adhesion. a** Biofilm adhesion assay for different Bap1 mutants (structure shown schematically) in a $\Delta rbmC$ strain background. BSA was used during biofilm growth at increasing concentrations as a non-specific competitor for glass surface adhesion. $\Delta BC$ denotes the $\Delta bap1\Delta rbmC$ double mutant. All data are depicted as the mean ± SD ($n = 3$ biologically independent samples). **b, c** In situ staining and associated quantification of biofilms formed by different Bap1 mutants, tagged by 3×FLAG at the C-terminus and labeled using anti-FLAG-Cy3. Note a functional copy of RbmC is present in this assay to anchor the biofilms to the substrate regardless of whether the mutant Bap1 is functional. **b** Representative cross-sectional images of the bottom layer and side views of biofilms formed by cells expressing WT Bap1 (*Left*), Bap1$_{\Delta57aa}$ (*Middle*), and Bap1$_{\Delta\beta\text{-prism}B+57aa}$ (*Right*). **c** Quantification of Bap1 localization within the biofilm for different mutants (structure shown schematically). Shown are the ratios between the immunosignals of 3×FLAG-tagged Bap1 at the biofilm-glass interface and the total signal integrated over the entire biofilm

cluster, for each indicated strain. Different colors and symbols correspond to different biological replicates ($n = 4$). Statistical analysis was performed using unpaired, two-tailed *t*-test with Welch's correction. ns stands for not significant; ***$p < 0.001$. Exact *p* values from left to right: 0.0007, 0.7163, 0.0003, 0.0665. **d** Peptide sequence of the 57aa loop. Blue dash = aromatic residues, red dot = positively charged residues. **e** Microbead adsorption assay for quantifying adhesive properties of the 57aa peptide. *Top*: a representative image of 5 μm silica bead (*Left*, bright field) and FITC-labeled 57aa peptide adsorbed on the bead (*Right*). *Bottom*: a representative image of 5 μm silica beads coated with lipids, labeled with RhPE (*Left*) for lipids and the FITC-labeled 57aa peptide adsorbed on lipid layer (*Right*). **f** Excess fluorescence signal on the surface of the beads compared to the solution signal, on silica surface (*Top*) and supported lipid layer (*Bottom*), respectively. FITC was used at the same molecular concentration (1.5 μM) as a control. **g** Confocal images of human jejunum tissue slices stained with 1 μM FITC-labeled 57aa peptide and 300 nm DAPI.

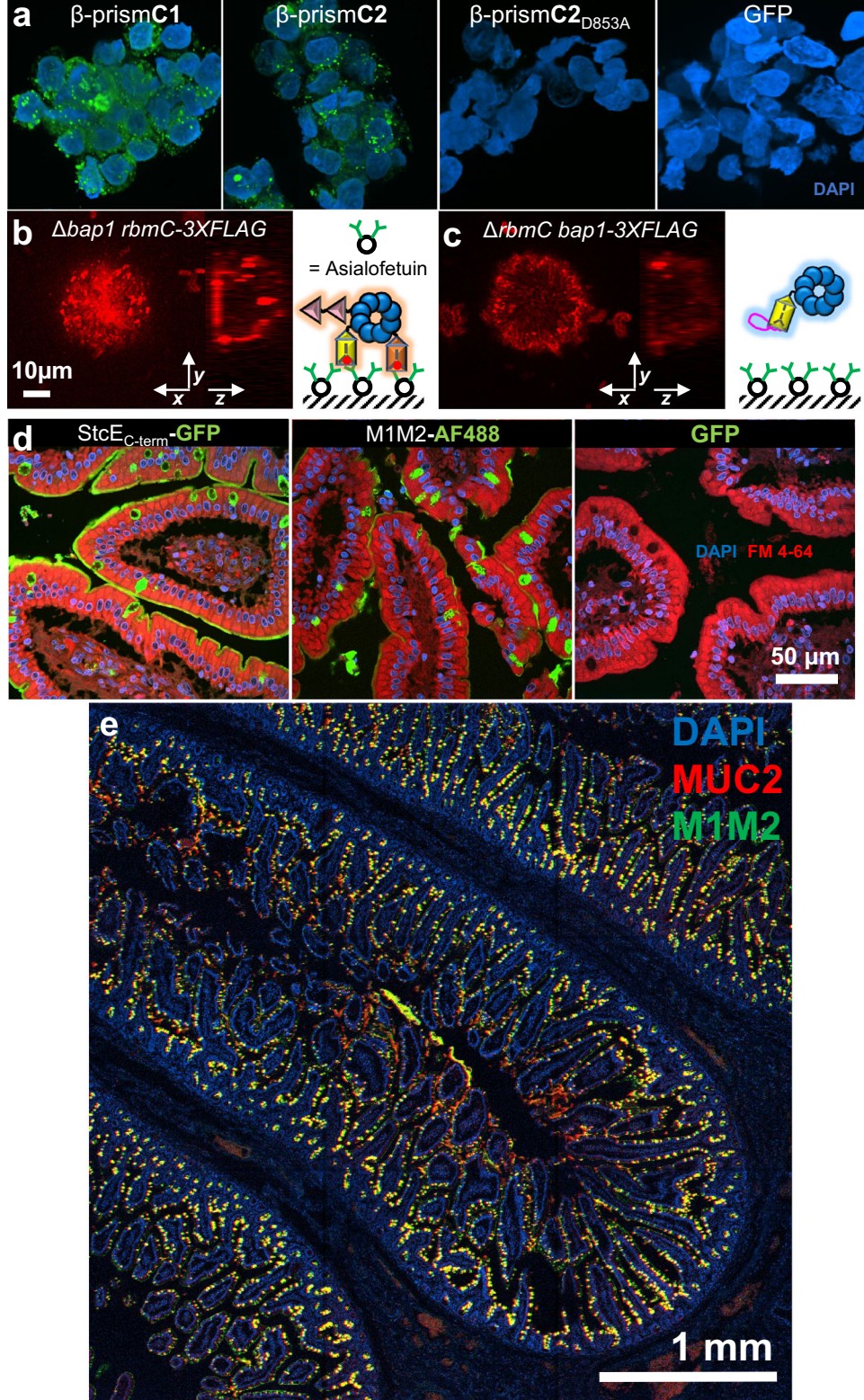

**Fig. 4 | RbmC targets host cell surfaces via N-glycan and mucin-binding domains. a** Merged z-stack of confocal images of DAPI-stained Caco-2 cells incubated with 1 μM purified and GFP-tagged β-prism domains. The total size of each image is 80 × 80 × 32.5 μm. **b, c** In situ immunostaining of RbmC (**b**) or Bap1 (**c**) in biofilms formed on an asialofetuin-coated glass surface. Shown on the right is the corresponding schematic representation of each protein's interaction with an asialofetuin-coated surface. **d** Confocal images of human jejunum tissue slices stained with DAPI, FM 4-64, and 1 μM purified and GFP-tagged StcE$_{C-term}$ from *E. coli* (*Left*), Alexa Fluor-488 conjugated M1M2 (M1M2-AF488) from *V. cholerae* RbmC (*Middle*) and free GFP (*Right*). **e** Large-scale confocal images of human jejunum tissue slices stained with DAPI (blue), M1M2-GFP (green), and anti-MUC2-antibody (red).

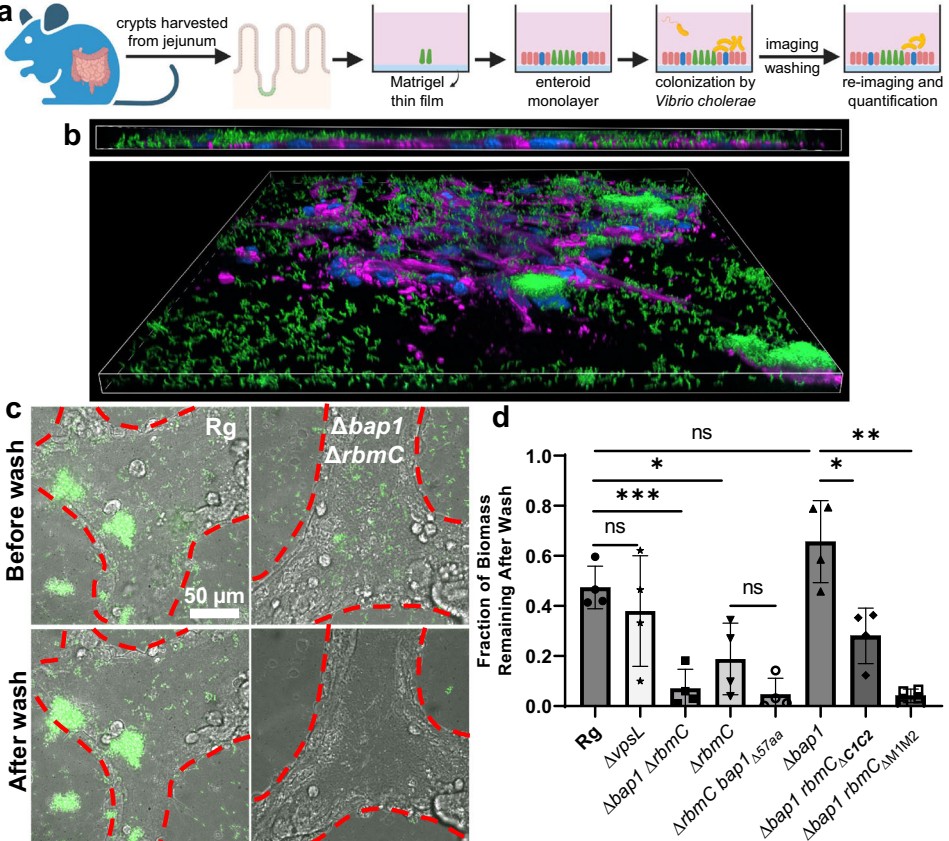

**Fig. 5 | Bap1 and RbmC are required for colonizing enteroid monolayers.**
**a** Schematic for the growth of in vitro enteroid monolayers and subsequent colonization by *V. cholerae* cells. Created with Biorender.com. **b** Side (*Top*) and tilted view (*Bottom*) of a representative monolayer stained with DAPI and an F-actin probe conjugated to fluorescent Alexa Fluor™ 647 dye (magenta) and colonized by *V. cholerae* biofilms from a rugose (Rg) strain constitutively expressing mNeonGreen. The total size of the view is $439 \times 439 \times 13\,\mu$m. **c** Representative results of a monolayer colonized by *V. cholerae* biofilms, before (*Top*) and after (*Bottom*) washing,

showing defective colonization of the $\Delta rbmC\Delta bap1$ mutant. Red dotted line demarks the boundary of the monolayer. **d** Quantification of monolayer colonization of different biofilm mutants (mean ± SD, $n = 4$ biologically independent samples, unpaired, two-tailed $t$-test with Welch's correction. *$p < 0.05$ (Rg v.s. $\Delta rbmC$: $p = 0.0193$; $\Delta bap1$ v.s. $\Delta bap1\ rbmC_{\Delta C1C2}$: $p = 0.0114$), **$p < 0.01$ ($\Delta bap1$ v.s. $\Delta bap1\ rbmC_{\Delta M1M2}$: $p = 0.0042$), ***$p < 0.001$ (Rg v.s. $\Delta bap1\Delta rbmC$: $p = 0.0004$), ns = not significant (Rg v.s. $\Delta vpsL$: $p = 0.4730$; Rg v.s. $\Delta bap1$: $p = 0.1107$; $\Delta rbmC$ v.s. $\Delta rbmC\ bap1_{\Delta57aa}$: $p = 0.1430$).

adhesins (RbmC and Bap1) is critical for the adhesion of *Vc* biofilms to the model intestinal epithelia (Fig. 5c), and that the separate domains of RbmC and Bap1 contribute differently to biofilm adhesion. Specifically, by calculating the ratio of biomass on the monolayer before and after washing, we first show that the double mutant abolishes adhesion to the enteroid monolayer. Moreover, a single deletion of *rbmC* but not *bap1* leads to a significant reduction in monolayer adhesion (Fig. 5d), suggesting that RbmC is the primary adhesin for colonizing enteroid monolayers (in contrast to results on abiotic surfaces). Furthermore, adhesion via *rbmC* depends on both the N-glycan targeting of the β-prism**C**s and the mucus-binding of M1M2, with the latter being the predominant factor (compare $\Delta bap1$, $\Delta bap1\ rbmC_{\Delta\text{-prism}C1C2}$, and $\Delta bap1\ rbmC_{\Delta M1M2}$ in Fig. 5d). Interestingly, the $\Delta vpsL$ mutant that makes no biofilm adheres at a comparable level to the parental strain. This indicates that other adhesive factors are functioning in the absence of VPS; however, when cells produce VPS in the biofilm state, assistance by biofilm-specific adhesins (mostly RbmC) is necessary to ensure proper adhesion to host surfaces.

## Bap1 and RbmC have overlapping function but distinct mechanisms and distributions in other Vibrio species
Integrating all the evidence presented above, we propose a model in which RbmC and Bap1 facilitate *V. cholerae* biofilm adhesion to divergent types of external surfaces with their non-conserved domains while sharing a conserved β-propeller domain that binds VPS, thereby

acting as unique bacterial "double-sided tape" (Fig. 1c, d). Specifically, we propose that Bap1 specializes in sticking to lipids and abiotic surfaces via a mussel-like chemistry while RbmC specializes in recognizing host surfaces via N-glycan-binding pockets in its β-prism**C**s and mucin-binding M1M2 domains. Consequently, *V. cholerae* biofilms can adhere to a wide range of foreign surfaces with different chemical properties. While most results were presented in the rugose background, we have confirmed the validity of our key conclusions in the WT background (Supplementary Fig. 8).

To put our findings into an evolutionary perspective, we performed bioinformatic analyses of Bap1 and RbmC homologs in the Vibrio genus. We found four other species that contain RbmC homologs (*V. tubiashii*, *V. coralliilyticus*, *V. anguillarum*, and *V. mimicus*) among the 21 species analyzed, with two other species showing weak partial hits (Supplementary Fig. 9a). One species, *V. anguillarum*, has both a RbmC homolog and a Bap1 homolog. Interestingly, the phylogenetic tree based on protein sequences (Supplementary Fig. 9b) shows a significant deviation from the species tree generated from the whole genome analysis[39], suggesting that Bap1 has likely evolved through a gene duplication event from RbmC and subsequently acquired new functionalities. This neofunctionalization idea is consistent with another interesting genomic feature: while RbmC (VC0930) is located between the two VPS biogenesis clusters along with the other matrix proteins, Bap1 (VC1888) is located outside these clusters and flanked by unrelated genes. The presence of RbmC and

Bap1 homologs in these *Vibrio* species suggests that they may use similar adhesion strategies to attach to their host in the marine environment by producing an exopolysaccharide similar to VPS.

## Discussion

In this paper, we use a combination of microscopy, bacterial genetics, and biochemical approaches to delve into the fundamentals of biofilm adhesion. We find that the biofilm adhesins in *V. cholerae* adopt a modular approach that, through evolution, acquired specialized functionalities to attach to diverse surfaces while maintaining their affinity for the native exopolysaccharide. This feature makes them biofilm-specific in contrast to classical adhesins that rely on direct anchoring on bacterial cell surface[40]. On the other hand, the two adhesins differ in the adhesive properties of accessory domains that interface with the external environment. Through this functional redundancy and diverse surface targeting, we suggest a strategy for how *V. cholerae* biofilms use multiple mechanisms to attach to both biotic and abiotic surfaces. It will be interesting to see if the double-sided-tape-like design is generalizable to other biofilms that rely on a synergy between extracellular proteins and polysaccharides.

Our results address many questions regarding *V. cholerae* biofilm adhesion but also present new ones. For example, the VPS-β-propeller binding might explain the close juxtaposition of RbmC and VPS signals in super-resolution microscopy[26]. It is likely that VPS-β-propeller binding is multivalent: previous data from mechanical measurements suggest crosslinking of VPS by RbmC/Bap1[41], which requires one RbmC/Bap1 molecule to bind to two or more VPS monomers. Further experiments are necessary to confirm this hypothesis. Another intriguing question arises as to whether VPS synthesized by one Vibrio species can be recognized by the β-propeller in another species; such crosstalk has been seen in autoinducer recognition during bacterial communication[42]. Also, it is interesting to compare our findings with those from the other matrix protein RbmA that binds VPS using FnIII domains[43,44], a completely different sugar-binding motif. Regarding interactions with abiotic surfaces through the 57aa loop, while we have tested various synthetic surfaces, an interesting question for further study is whether the conclusions can be extended to chitinous surfaces that *V. cholerae* often colonizes in the oceanic environment[10,17].

The structural homology and similar functions of RbmC$_{M1M2}$ and StcE$_{C-term}$ hint at a broad molecular strategy of mucus interactions. A BLAST search using the StcE$_{C-term}$ sequence reveals not only its presence in RbmC of other Vibrio species, many of which colonize and infect marine animals with exposed mucus layers, but also in a range of infectious species including Shigella, Salmonella, Streptococcus, and Listeria. This suggests that this domain may be a broadly utilized, yet not-well-characterized mucin-targeting mechanism in multiple pathogens[36]. Future structural, mutagenesis, and glycobiology work is needed to illuminate the exact glycopeptide epitope recognized by this mucus-binding domain.

In addition to the enteroid colonization assay, we also performed small intestinal colonization assays in infant mice;[45] however, consistent with prior literature[13,21], we did not find a statistically significant colonization defect of the Δ*bap1*Δ*rbmC* mutant compared to WT. Whole-animal models are known to be imperfect for studying *V. cholerae* adhesins due to the dominant role of toxin-coregulated pili (TCP) in colonizing the host; TCP often masks the contribution of other adhesive factors, which have been shown to be important for host adhesion in cell culture systems[46,47]. Future work on visualizing the spatial distribution of the different mutants in mice may address this discrepancy. The successful colonization of the Δ*vpsL* mutant on enteroid monolayers is consistent with the involvement of other adhesive factors such as GbpA[15,48] and OmpU[46]. However, our results emphasize that the colonization of the biofilm population relies critically on Bap1 and RbmC, because the VPS molecules surrounding the biofilm lack an adhesive property and may at the same time physically obstruct other adhesins from functioning. Because colonization of *V. cholerae* involves both planktonic and biofilm populations and the latter population is more infectious[12,49], our results suggest that biofilm-specific adhesins can work synergistically with well-known classical adhesins to contribute to *V. cholerae* pathogenicity.

## Methods

### Bacterial strains

All *V. cholerae* strains used in this study were derivatives of the WT *V. cholerae* O1 biovar El Tor strain C6706str2 and listed in Supplementary Table 1. The rugose strain background harbors a missense mutation in the *vpvC* gene (*vpvC*$^{W240R}$) that elevates intracellular c-di-GMP levels[30]. The rugose strains form robust biofilms and thus allow us to focus on the biochemical mechanisms governing biofilm adhesion rather than mechanisms involving gene regulation. Additional mutations were genetically engineered into this *V. cholerae* strain using the natural transformation (MuGENT) method[50].

### Bacterial growth

All strains were grown overnight in lysogenic broth (LB) at 37 °C with shaking. 1× M9 salts were filter sterilized and supplemented with 2 mM MgSO$_4$ and 100 µM CaCl$_2$ (abbreviated as M9 medium below). Biofilm growth was generally performed in M9 medium supplemented with 0.5% glucose. For complementation experiments, 100 µg/mL Kanamycin was used.

### Strain construction

Linear PCR products were constructed using splicing-by-overlap extension (SOE) PCR as previously described and used as transforming DNA (tDNA) in chitin-dependent transformation reactions[50]. Briefly, SOE PCR was performed by amplifying an upstream region of homology and a downstream region of homology. The desired mutations were incorporated into the primers used in amplification. All primers used to construct and detect mutant alleles are listed in Supplementary Table 2. For chitin-dependent transformation, individual *V. cholerae* colonies were grown in LB media at 30 °C for 6 h to an OD$_{600}$ = 0.8–1.0. Cells were washed with Instant Ocean (IO) solution and then incubated with chitin particles suspended in IO for 8–16 h at 30 °C before the tDNA was added. The cultures were then incubated at 30 °C for an additional 8–16 h. LB was added to the cultures and incubated at 37 °C for 2 h before plating on LB agar with the appropriate antibiotic. The desired mutants were selected by the emergence of new phenotype or colony PCR screening and confirmed by sequencing and complementation.

### Biofilm adhesion assay

Overnight cultures of the indicated strains constitutively expressing mNeonGreen were grown from individual colonies at 37 °C with shaking in 1.5 mL LB. 50 µL from each culture was used to inoculate 1.5 mL of M9 medium supplemented with 0.5% glucose and grown at 30 °C with shaking until the OD$_{600}$ was between 0.1 and 0.3. The cultures were then diluted to an OD$_{600}$ ≅ 0.001. 100 µL of the regrown culture was aliquoted into the wells of a 96-well plate with a glass bottom (MatTek P96G-1.5-5-F) and incubated at 30 °C for 1 h. The wells were then washed twice with M9 medium and replaced with M9 medium with 0.5% glucose and 0, 0.2, 0.4, 0.6, 0.8, 1, or 1.2 mg/mL BSA. In the complementation experiment, the growth medium additionally contains 0.2% arabinose for P$_{BAD}$-*bap1*. The lid was secured with a layer of parafilm and the 96-well plate was subsequently incubated at 30 °C for 16–24 h. Thus-prepared samples were imaged with a spinning disk confocal microscope (Nikon Ti2-E connected to Yokogawa W1) using a 60× water objective (numerical aperture = 1.20) and a 488 nm laser excitation. For each sample, several locations with 3 × 3 tiles where imaged and captured with a sCMOS camera (Photometrics Prime BSI). The *x*-*y* pixel size was 0.22 µm and the *z*-step size was 3 µm. The wells

were then washed twice with M9 medium and re-imaged at the same locations. All images presented in this study are raw data rendered using the Nikon Elements software.

### NaOH treatment of glass substrates

For a subset of experiments (Supplementary Fig. 3b,d), the surface of the 96-well plate was treated with NaOH to render it more hydrophilic and negatively charged[51]. Briefly, before adding the cell culture, 100 μL of 1 M NaOH aqueous solution was added to the wells and incubated at room temperature for 3 h, after which the wells were washed with DI water until the pH was neutral.

### Quantification of adhesion assays

Image analysis was performed with built-in functions of the Nikon Elements software by thresholding each image layer-by-layer and measuring the total binarized area above the threshold in each layer. The binary area for each sample $z$-slice was then summed to give the total biovolume, and the ratio of the total biovolume after versus before the washing step was calculated.

### Growth of WT *V. cholerae* biofilms

To verify our findings in the WT background, we generated all Bap1 mutants in the WT C6706 background. All strains were grown in LB medium at 30 °C with shaking overnight. 50 μL from each culture was used to inoculate 1.5 mL of M9 medium supplemented with 0.5% glucose and 0.5% casamino acids and grown at 30 °C with shaking until the $OD_{600}$ was between 0.1 and 0.3. Growth of WT *V. cholerae* biofilms was performed according to a published protocol with modifications[52]. The inoculants previously described were introduced into microfluidic channels (channel dimensions: 1 cm in length, 400 μm in width, and 60 μm in height) through the outlet without inoculating the inlet. The cells were allowed 1 h to attach, after which sterile inlet and outlet polytetrafluoroethylene tubing was connected to the microfluidic chamber. M9 medium supplemented with 0.5% glucose and 0.5% casamino acids, with or without 0.4 mg/mL BSA, was flowed through the channel at a flow rate of 0.6 μL/min controlled by a syringe pump. After 16 h growth at 30 °C, the microfluidic channels were transferred to a spinning disk confocal microscope, with which the biomass was imaged and quantified using procedures similar to those described above. One exception was that the cell layer directly adhered to the glass surface was excluded from the analysis, because cells in this layer mostly attach to surfaces as individual cells via pili rather than as biofilms[53,54].

### Biofilm staining with purified proteins

*V. cholerae* biofilms from cells constitutively expressing mScarlet-I were grown as described above. After overnight biofilm growth, the growth media was replaced with 100 μL of M9 media containing 1 μM of purified GFP-tagged Bap1 domain constructs. The samples were incubated for 30 min at room temperature. The media containing protein was then removed and replaced with fresh M9 media. The samples were imaged with a spinning disk confocal microscope using a 60× water objective and a 488 nm laser excitation to observe protein localization and a 561 nm laser excitation to observe the biofilm, with the corresponding filters.

### In situ biofilm immunostaining

Overnight cultures of the indicated strains with WT or mutated *rbmC* or *bap1* tagged with 3×FLAG at the C-terminus and constitutively expressing mNeonGreen were grown following the same procedure as described above. The initial incubation time was 10 or 30 min when biofilms were grown in the presence of asialofetuin or BSA, respectively. The wells were washed twice with M9 medium; subsequently, 100 μL of M9 medium with 0.5% glucose and 1 mg/ml asialofetuin (Sigma-Aldrich A4781) or 0.5 mg/ml BSA (Sigma-Aldrich A9647) was added to the well.

BSA and asialofetuin spontaneously coat the surface under these conditions. Both conditions included 2 μg/mL anti-FLAG antibody conjugated to Cy3 (Sigma-Aldrich A9594). The lid was secured with a layer of parafilm and incubated at 30 °C for 16–24 h for asialofetuin and 40–48 h for BSA samples. Thus-prepared samples were imaged with a spinning disk confocal microscope (Nikon Ti2-E connected to Yokogawa W1) using a 100× oil immersion objective (numerical aperture = 1.35) or a 60× water immersion objective (numerical aperture = 1.20) and a 488 nm laser excitation to observe the cells and a 561 nm laser excitation to observe protein localization, with the corresponding filters. The images were captured with a sCMOS camera (Photometrics Prime BSI) at a $z$-step size of 0.5 μm.

### Quantification of protein distribution in biofilms

We used the in situ immunostaining image stacks for quantifying protein distribution using built-in functions of the Nikon Element software. First, background noise in the 561 nm channel was measured by taking images in locations without any biofilms and subtracted from the data. Next, a circular region of interest was manually defined that contains a single biofilm cluster. To be consistent, we only included biofilms of similar heights (25–30 μm) in the analysis. Subsequently, anti-FLAG-Cy3 signals at the glass surface ± 0.5 μm were added and the total signal over the entire biofilm height was integrated; the ratio between the two values was calculated to quantify the ability of the adhesin to preferentially localize at the biofilm-glass interface. The sizes of the puncta in the *bap1*$_{\Delta Velcro}$ biofilms (Supplementary Fig. 2h) were manually measured using built-in tools in Nikon Element Software.

### Crystal violet assay

The indicated *V. cholerae* strains were grown on LB agar plates at 37 °C, and individual colonies were picked to inoculate culture tubes with 3 mL LB and glass beads. The cultures were grown at 37 °C with shaking until exponential phase ($OD_{600}$-0.5). 1 × 3-inch glass slides were cut into similar sizes, washed with ethanol, and flame sterilized before being inserted into sterile culture tubes containing 1 mL LB. Exponential phase cultures were used to inoculate the cell culture tubes with the glass slides at an $OD_{600} = 0.01$ (for example, 20 μL of a culture at $OD_{600} = 0.5$ was used to inoculate the tube containing 1.0 mL LB and a glass slide). The cultures were grown statically at 37 °C for 16 h. One at a time, the glass slides were carefully removed and washed 3 times with DI water, stained with 1.5 mL of a 0.1% crystal violet solution for 10 min, washed 3 times with DI water, and transferred to a fresh tube containing 1.5 mL of 30% acetic acid to dissolve the stain associated with the pellicles. The stained acetic acid solution was then transferred to a 1.5 mL cuvette to measure the $OD_{550}$.

### Western blots and secretion assay

*V. cholerae* strains encoding the indicated constructs with a C-terminal 3×FLAG tag were grown in culture tubes containing 3 mL LB and sterile glass beads overnight at 30 °C. The next day, cultures were vortexed to break up pellicles and cell clusters and the $OD_{600}$ was measured. 1 mL of cell suspensions were transferred to a sterile 1.5 mL microcentrifuge tube and spun at 18,000 × g for 3 min. 500 μL of the cell supernatant was transferred to a fresh 1.5 mL microcentrifuge tube and the rest discarded from the pellet. The cell pellets were resuspended to an $OD_{600} = 10$ and lysed for 30 min using a lysis solution (1× Bugbuster solution, lysozyme (0.05-0.1 mg/mL), and benzonase (≥ 250 units/mL)). 30 μL of each cell suspension was combined with 10 μL of 4× SDS PAGE sample buffer (40% Glycerol, 240 mM Tris pH 6.8, 8% SDS, 0.04% Bromophenol Blue, 5% β-mercaptoethanol) and boiled for 10 min at 95 °C. Samples were run on a 4–15% Mini-PROTEAN TGX gel in 1× SDS PAGE running buffer (25 mM Tris, 192 mM Glycine, 1% SDS, pH 8.3) at 120 V for 70 min. The proteins were transferred to a PVDF membrane in 1× Transfer buffer (25 mM Tris, 192 mM Glycine, 10% methanol, pH

8.3) at 100 V for 1 h. The membranes were incubated in 5% milk in TBST overnight at 4 °C. The membranes were washed 3 × 10 min in 1× TBST. The membranes were blotted using α-DYKDDDDK (BioLegend 637311) at 0.1 µg/mL in TBST with 3% BSA for 1 h at room temperature and washed 3 × 10 min with 1× TBST. Blots were developed by incubation with Super Signal PLUS Pico West Chemiluminescent Substrate for 5 min and pictures taken using the BioRad Chemidoc-MP. Analysis of sample signal was performed in ImageJ.

### *E. coli* protein expression and purification

GFP$_{UV}$-tagged proteins were cloned, expressed, and purified in *E. coli*[27,28] with the exception of new clones listed in Supplementary Table 1. New constructs were made by PCR from *V. cholerae* genomic DNA or previously cloned genes using primers listed in Supplementary Table 2 and traditional sticky-end cloning into the GFP$_{UV}$ fusion vector pNGFP-BC[55]. RbmC$_{\beta-propeller}$ was made using a two-step PCR stitching reaction with primers listed in Supplementary Table 2. Clones containing inserts were confirmed by DNA sequencing. For expression and purification, LB media supplemented with 100 µg/mL carbenicillin was inoculated with overnight cultures grown at 37 °C to an OD$_{600}$ of 0.5–0.6, induced with 1 mM IPTG, and grown at 18 °C overnight. Cells were pelleted at 5462 × g in a Sorvall LYNX 6000 centrifuge (F9-6×1000 LEX rotor) for 15 min and lysed by passing three times through an Emulsiflex-C5 high-pressure homogenizer (Avestin, Inc.). Lysate was cleared at 41656 × g for 30 min at 4 °C (F20-12 × 50 LEX rotor). The resulting supernatant was loaded onto a 5 mL HisTrap Ni-NTA column (GE Healthcare) equilibrated in 1× TBS (20 mM Tris-HCl, pH 7.8, 150 mM NaCl) and washed with 1× TBS containing 40 mM imidazole. Protein was eluted with 15 mL of 1× TBS containing 250 mM imidazole. Protein samples were further purified over a Sepharose S6 10/300 size-exclusion column (GE Healthcare) preequilibrated with 1× TBS. Protein fractions were pooled after assessing purity using an SDS-PAGE gel.

### Fluorescent labeling of RbmC$_{M1M2}$

Purified RbmC$_{M1M2}$ was fluorescently labeled by primary-amine chemistry using an Alexa Fluor 488 TFP ester reagent (Thermo Scientific A37570). Prior to labeling, purified RbmC$_{M1M2}$ was buffer exchanged into 1× phosphate-buffered ½ saline (10 mM Na$_2$HPO$_4$, 1.8 mM KH$_2$PO$_4$, 2.7 mM KCl, and 75 mM NaCl, pH 8.3). For labeling, 1 mg/mL of RbmC$_{M1M2}$ was used. While stirring, 100 µg of the dye (resuspended in 10 µL DMSO) was added to 600 µL of protein-containing solution and incubated for 1 h at room temperature. Unreacted dye was removed from labeled proteins by running over a Superose 6 10/300 size exclusion column equilibrated with 1× phosphate-buffered ½ saline.

### Caco-2 cell culturing and staining

Human colonic epithelial Caco-2 cells (ATCC HTB-37) were obtained from ATCC and authenticated by ATCC based on morphology, doubling time, and STR profiling. Caco-2 cells were cultured in flasks containing Dulbecco's Modified Eagle's Medium (DMEM; Gibco) supplemented with 10% (v/v) heat-inactivated fetal bovine serum (FBS-HI; Gibco) at 37 °C in a humidified 5% CO$_2$ incubator. After 72 h, cells were collected via dissociation using TrypLE Express (Gibco) and pelleted by centrifugation (300 rcf, 3 min, room temperature in 15 mL conical tubes (Corning); then 21,000 rcf, 2 min, room temperature in Eppendorf tubes). Cell pellets were stored at −80 °C prior to further analysis.

To stain Caco-2 cells with purified proteins, a frozen aliquot of Caco-2 cells as prepared above was gently thawed and then added to 1 mL of M9 medium containing 300 nM DAPI and incubated for 5 min at room temperature. 100 µL of this cell suspension was aliquoted to sterile 1.5 mL microcentrifuge tubes and spun at 10,000 × g for 3 min. The staining media were removed and replaced with 100 µL of M9 media containing 1 mg/mL BSA and 1 µM of purified GFP-tagged Bap1/RbmC β-prism domain constructs or GFP alone. The samples were

incubated for 30 min at room temperature and then the media was replaced with 100 µL fresh M9 medium and transferred to the wells of a 96-well plate. The samples were imaged with a spinning disk confocal microscope using a 60× water objective and a 405 nm laser excitation to observe the Caco-2 cell nuclei and a 488 nm laser excitation to observe protein localization, with the corresponding filters.

### Mice

All animal work was approved by Yale University's Institutional Animal Care and Use Committee. CD1 (Charles River) mice of both sexes, aged 4-12 weeks were used in this study. CD1 IGS mice were purchased from Charles River Laboratories, Strain 022 and were bred for up to two generations within the Yale Animal Resource Center. Mice were maintained in ventilated Techniplast limit racks with ambient temperature of 22 °C and 50%±10% humidity in a barrier facility with 12 h light/dark cycles. They were given ad libitum access to food and water.

### Enteroid monolayer generation and culture

96-well Black/clear plates (Corning 353219) were coated with 30 µL growth factor reduced Matrigel (Corning 356231) diluted 1:5 in Basal organoid medium. Plates were incubated at 37 °C for at least 1 h to allow Matrigel to polymerize. Basal organoid medium was comprised of: advanced DMEM/F12 (Thermo Fisher 12634010) supplemented with 1× N-2 supplement (Thermo Fisher 17502-048), 1× B-27 supplement (Thermo Fisher 17504044), 10 mM HEPES (AmericanBio AB6021-00100), 1× Glutamax (Thermo Fisher 35050061), 1 mM N-acetyl-cysteine (Sigma Aldrich A9165), and 1× Penicillin/Streptomycin (Thermo Fisher 15140-122).

Enteroid monolayers were generated as described previously[38] with modifications. Briefly, ~4 cm of jejunum was removed from 4 to 12 week old mice, flushed with ice-cold PBS and cut open longitudinally to expose the epithelium. The tissue was scraped with a 22 × 22 mm coverslip to remove villi and placed in PBS + 3 mM EDTA at 4 °C with rotation for 30 min. The tissue was then manually shaken with forceps in a 6 cm petri dish to release villi. The PBS was replaced to deplete villar fractions. This process was repeated until the PBS contained mostly crypts by visual inspection. The solution containing crypts was strained through a 70 µm filter (Fisher) and centrifuged at 300 × g for 3 min to pellet crypts. Crypts were resuspended in 2D attachment media, which consisted of basal organoid media (above) supplemented with 50 ng/mL EGF (Thermo Fisher PMG8041), 100 nM LDN-193189 (Cayman, 11802), 1 µg/mL R-spondin 1 (R&D Systems, 3474-RS-050), 10 µM CHIR99021 (Cayman, 13122), and 10 µM Y27632 (Tocris, 1254). 100 µL of resuspended crypts were added to each Matrigel-coated well of a 96-well plate and incubated at 37 °C for 4 h. The wells were then washed 3 times with PBS and placed in supplemented ENR media for the remainder of the culture. Supplemented ENR media was comprised of basal organoid media plus 50 ng/mL EGF, 50 ng/mL Noggin (R&D Systems, 6057-NG-100), and 1 µg/mL R-spondin 1. Media was replaced every other day and put into antibiotic-free media the morning of the bacterial colonization experiments.

### Staining and visualization of fixed monolayers

After four days of culture, monolayers were fixed for 10 min in 4% paraformaldehyde (PFA). After washing in 1× PBS, monolayers were incubated in blocking buffer (3% BSA, 5% NDS, 5% NGS, 0.2% Triton X-100 in PBS) for 45 min. Cells were incubated in primary antibody diluted in blocking buffer for 2 h at RT. Wells were washed with 1× PBS + 0.2% Triton X-100 and incubated for 5 min. This step was repeated three times. Secondary antibodies were diluted in blocking buffer and incubated on monolayers for 45 min. Primary antibodies used were as follows: CD44v6 1:100 (Invitrogen, BMS145), Villin 1:400 (BD Biosciences, 610358), Muc2 1:1000 (Abcam, ab272692). Secondary antibodies were all used at 1:200: Rhodamine Red X D anti Rt (Jackson Immunoresearch 712-295-153), Alexa Fluor 488 D anti Rb (Jackson

Immunoresearch, 711-545-152), Alexa 647 D anti Rb (Jackson Immunoresearch, 711-605-152). DAPI was used 1:500 (Thermo Fisher, D1306). Stained monolayers were stored in 1× PBS and imaged on an inverted Leica Stellaris 5 using Leica LASX Version 4.3.0.24308 with white light laser using a 25×/0.95 HC Fluotar water immersion objective.

For protein staining, the samples were washed once with 1× PBS. After the wash, the samples were fixed with pre-warmed 4% PFA in 1× PBS for 10 min. The fixation solution was removed, and the samples incubated in 1× PBST containing 0.2% Triton X-100 for 10 min at room temperature. Monolayers were then blocked in 1% BSA in 1× PBS for 30 min with shaking, before being incubated in staining solution containing 300 nM DAPI, 1 mg/mL BSA, 0.66 μM Alexa Fluor 647 phalloidin (Invitrogen, A22287) and 1 μM of purified protein in 1× PBS. The monolayers were washed with 1× PBS twice before imaging. The samples were imaged with a spinning disk confocal microscope using a 60× water objective and a 405 nm laser excitation to observe monolayer nuclei, a 488 nm laser excitation to observe protein localization, and a 647 nm laser excitation to observe actin, with the corresponding filters.

## Colonization of enteroid monolayers

The indicated *V. cholerae* strains constitutively expressing mNeon-Green were grown overnight at 30 °C on LB agar with the appropriate antibiotic. An isolated colony from each strain was used to inoculate 1.5 mL of LB with glass beads (4 mm, MP Biomedical) and grown with shaking for 16–18 h at 30 °C. For each strain, small glass beads (acid-washed, 425–500 μm, Sigma) were added to a 1.5 mL microcentrifuge tube up to the 100 μL line. 30 μL of each culture was added to 1.5 mL of M9 medium supplemented with 0.5% glucose and vortexed to mix. 500 μL of the culture was added to each microcentrifuge tube with acid-washed beads and grown with shaking for 16–18 h at 30 °C. The overnight culture was bead bashed using a Disruption Genie for 10 min at 3000 rpm. Once bashed, the cultures were left on the bench for 5 min to allow the beads to settle. The bashed liquid culture was then removed from the top of the beads and added to a fresh 1.5 mL microcentrifuge tube. The bead-bashing step is necessary to break up large biofilm clusters that may have emerged during overnight growth. Using the separated culture, the $OD_{600}$ for each strain was measured. The remaining overnight culture was pelleted at $18,000 \times g$ for 90 s. The cell supernatant was removed before the cells were resuspended to the original $OD_{600}$ in pre-warmed DMEM. The resuspended cultures were diluted for colonization to an $OD_{600} = 0.3$ in 100 μL of DMEM.

To colonize the monolayers, the supernatant was carefully removed from each well with the selected monolayer and the bacterial culture was added slowly into the well. The bacteria were allowed to be in contact with the monolayers at 37 °C for 60–75 min to establish adhesion. Once the incubation period was complete, the samples were imaged with a spinning disk confocal microscope using a 60× water objective (N.A. = 1.20) and 488 nm laser excitation to observe bacterial localization and a brightfield camera to observe monolayer boundaries, with the corresponding filters. The samples were then carefully washed twice with pre-warmed DMEM and re-imaged at the same locations. Quantification of adhesion to monolayers was performed in a procedure similar to described above, with several modifications: The boundary of the monolayer was manually traced for each imaged area, and only biomass within the boundary was used for adhesion quantification. Additionally, the largest biofilm cluster was excluded from quantification for each sample to prevent large, floating clusters from artificially decreasing the adhesion quantification for each strain.

For actin staining of colonized monolayers, the monolayers were incubated for 30 min at room temperature with rocking in a staining solution of 1 μM DAPI and 0.66 μM Alexa Fluor 647 phalloidin (Invitrogen, A22287) in 1×PBS with 1 mg/mL BSA. Monolayers were carefully washed with 1×PBS twice. The samples were imaged with a spinning disk confocal microscope using a 60× water objective and a 405 nm

laser excitation to observe monolayer nuclei, a 488 nm laser excitation to observe bacterial localization, and a 647 nm laser excitation to observe actin, with the corresponding filters.

## Staining and visualization of jejunum tissue slices

Pre-fixed human jejunum tissue slices were obtained from Novus Biologicals (NBP2-30201). Prepared slides were deparaffinized according to the manufacturers protocol. Briefly, the slides were dried for 1 h at 60 °C and then soaked in xylene 5 × 4 min. The slides were then hydrated in 100%, 95%, and 75% ethanol 2 × 3 min and immersed in water for 5 min. Staining solutions containing 4 μg/mL FM 4-64, 300 nM DAPI, 1 mg/mL BSA and 1 μM of purified protein in 1×PBS were added to the slides and incubated for 30 min at room temperature. Slides were carefully washed twice with 1×PBS. The samples were imaged with a spinning disk confocal microscope using a 60× water objective and a 405 nm laser excitation to observe the intestinal cells' nuclei, a 561 nm laser excitation to observed cell membranes, and a 488 nm laser excitation to observe the protein localization, with the corresponding filters.

For antibody staining, the slides were incubated with shaking in 1×PBS with 1 mg/mL BSA for 30 min at room-temperature. A staining solution of diluted primary antibody in 1×PBS with 1 mg/mL BSA was then added to the slides and slides were incubated with rocking for 1 h at room temperature. Slides were carefully washed three times with 1×PBS for 5 min each. After washing, the slides were incubated with rocking in a staining solution of diluted secondary antibody in 1×PBS with 1 mg/mL BSA for 1 h at room temperature. Slides were carefully washed three times with 1×PBS for 5 min each. Slides were then stained with relevant proteins following the protocol previously described. The samples were imaged with a spinning disk confocal microscope using a 10× objective and a 405 nm laser excitation to observe intestinal cells' nuclei, a 488 nm laser excitation to observe M1M2-GFP localization, and a 647 nm laser excitation to observe MUC2 localization, with the corresponding filters.

## Microbead adsorption assay

Chemically synthesized peptides (Atlantic Peptides) were dissolved and stored in DMSO at 150 μM and diluted 100× into M9 media immediately before the adsorption assay. 100 μL of M9 media containing 1.5 μM of FITC-labeled peptide or FITC and 0.01% (weight percent) 5 μm silica microspheres (Polysciences 25348) was shaken in Eppendorf tubes for 30 min at room temperature. The sample was then bath sonicated for 20 min with ice before transferring to a NaOH-treated 96-well plate with a glass bottom (MatTek P96G-1.5-5-F) and allowed to settle at room temperature for 5 min before imaging. Thus-prepared samples were imaged with a spinning disk confocal microscope (Nikon Ti2-E connected to Yokogawa W1) using a 60× oil objective (numerical aperture = 1.40) and a 488 nm laser excitation or bright field. For each sample, at least three locations were imaged and captured with a sCMOS camera (Photometrics Prime BSI). Each field of view contained roughly 100–150 beads.

## Lipid-coated microbead adhesion assay

Silica microbeads were coated with lipid layers according to published protocols with modification[56]. Briefly, 75 mol% PC (Avanti Polar Lipids 840051), 25 mol% PI (Avanti Polar Lipids 840042), and >0.1 mol% L-α-phosphatidylethanolamine-N-(lissamine rhodamine B sulfonyl) (abbreviated as RhPE, Avanti Polar Lipids 810146) were mixed in chloroform in a glass vial prerinsed with chloroform. A light stream of nitrogen was used to remove excess solvent, followed by at least 2 h in a vacuum desiccator. Lipids were hydrated for 30 min at 37 °C at a final lipid concentration of 5 mM in buffer (20 mM Tris, pH 8.0, 300 mM KCl, and 1 mM $MgCl_2$) with vortexing and agitation roughly every 5 min and probe sonicated to clarity (4 min, with intermittent breaks) to form small unilamellar vesicles (SUVs). SUVs were adsorbed onto 5 μm silica microspheres by mixing 50 nmol lipids with 440 mm² of silica

microspheres surface area in a final volume of 80 μL and 1 h rotary shaking at room temperature. Excess SUVs were removed by pelleting coated beads for 30 s at 862 × g followed by washing 4 times with excess buffer (100 mM KCl and 50 mM Tris, pH 8.0). 100 μL buffer (100 mM KCl, 50 mM Tris, pH 8.0, 0.1% methylcellulose (Sigma-Aldrich M7027), 0.1% BSA) with 1.5 μM of FITC-labeled peptide and 0.01% lipid-coated beads was transferred to a NaOH-treated 96-well plate with a glass bottom and incubated for at least 1 h at room temperature. Thus-prepared samples were imaged with a spinning disk confocal micro-scope (Nikon Ti2-E connected to Yokogawa W1) using a 60× water objective and a 488 nm laser excitation or a 561 nm laser excitation. For each sample, at least three locations were imaged and captured with a sCMOS camera (Photometrics Prime BSI). Each field of view contained roughly 100–150 beads.

### Quantification of bead adsorption assay

The background signal due to the camera in the 488 nm channel was measured by taking images of M9 medium and quantifying it with built-in functions of the Nikon Element software. After subtracting the background signal, the signal intensity per unit area in the adsorption layer of the beads and in the solution was calculated using MATLAB and the difference was determined to give the excess surface signal.

### VPS purification

VPS purification was performed according to a published protocol with several modifications[20]. First, a rugose ΔrbmAΔbap1ΔrbmCΔpomA strain was used as the starting strain for easier separation of cells and VPS and to avoid confounding factors due to matrix proteins. This strain was grown in LB at 30 °C overnight. 50 μL of this inoculum was added into 3 mL of LB liquid medium containing glass beads, and the cultures were grown with shaking at 30 °C for 3–3.5 h. 50 μL of this inoculum was applied to an agar plate con-taining M9 medium with 0.5% glucose and 0.5% casamino acids and shaken with glass beads to enable growth of a biofilm covering the entire plate. Plates were incubated at 30 °C for 2 days to form a continous bacterial lawn. For each purification batch, 10 plates were used. The biofilms were scraped off the agar plates carefully and resuspended in 1× PBS. Biofilm cells were collected by centrifugation (5000 × g, 4 °C, 45 min). The supernatant was clarified with addi-tional centrifugation (8000 × g, 4 °C, 45 min) and dialyzed for 2 days against distilled water using a dialysis cassette (10 kDa MWCO) with repeated water changes. The dialyzed sample was lyophilized to prepare crude VPS extract. The crude extract was dissolved in 10 mM Tris buffer at 1.5 mg/mL, treated with DNAse and RNAse (37 °C, 24 h), and then Proteinase K (37 °C, 48 h), followed by ultracentrifugation at 100,000 × g for 1 h to remove lipopolysaccharide. This solution was dialyzed against water for 3 days and lyophilized to provide VPS for the binding assay. For each purification batch, typically 10 mg of VPS was obtained as a white powder after the final lyophilization step. The VPS solutions were heated at 95 °C for 10 min to denature Proteinase K before use.

### VPS or BSM binding assays

Gels were prepared with a final concentration of 10% acrylamide (Bio-Rad) in the running gel and 5% in the stacking gel. The native running buffer contained 25 mM Tris-HCl, pH 8.3, and 192 mM glycine. The native loading buffer was made with 62.5 mM Tris-HCl, pH 6.8, 25% glycerol and 1% bromophenol blue dye. Samples for the gel-shift assay were prepared with 5 μg of protein per sample. For the VPS gradient, 0, 0.0625, 0.125, 0.25, 0.50, 1, and 5 μg of VPS was pre-incubated with the representative protein for 5 min. For the BSM gradient, 0, 0.5, 1, 2, 4, 6, 8, and 10 μg of BSM was preincubated with RbmC$_{M1M2}$ for 5 min. For the GFP control, the highest amount of VPS or BSM was used. For testing different polysaccharides, 5 μg (highest amount used in the VPS concentration gradients) was preincubated

with the Bap1$_{Δ57aa}$ (5 μg) for 5 min. Gel electrophoresis was per-formed at 85 V for 4 h in an ice bath. Images were acquired on gels (still encapsulated in glass) with an excitation wavelength of 492 nm and an emission wavelength of 513 nm using a Typhoon FLA 9000 imaging system (GE Healthcare).

### Phylogeny analysis

Bap1 or RbmC protein sequences from V. cholerae were used as a query to BLAST against each of the 20 genomes of Vibrio species[39]. BLAST hits with an E-value lower than 1e$^{-15}$ and alignment coverage (fraction of overlapping positions over the sequence alignment length) higher than 80% of the query were recorded as significant hits - the query gene is recovered in the target genome. Blast hits with an E value higher than 1e$^{-15}$ but lower than 1e$^{-5}$ were recorded as potential hits of the query gene with low conservation. A protein sequence alignment containing significant hits and the query sequence was manually examined to confirm the recovery of the query gene. The Bap1 protein with a 6aa insertion in V. cholerae O16 str. 877-163 was recovered from a BLAST search against NCBI nr database.

### Statistics and reproducibility

Error bars correspond to standard deviations from measurements taken from distinct samples. Standard t-tests were used to compare treatment groups and are indicated in each figure legend. Tests were always two-tailed, unpaired, and used Welch's correction, as deman-ded by the details of the experimental design. All statistical analyses were performed using GraphPad Prism software. Microscopy images were shown from representative results from at least three indepen-dent experiments.

### Reporting summary

Further information on research design is available in the Nature Portfolio Reporting Summary linked to this article.

## Data availability

Source data are provided with this paper.

## Materials availability

All bacterial strains constructed as part of this work will be provided to the community upon request in a timely fashion and shipped in accordance with biosafety standards and regulations.

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

## Acknowledgements

J.Y. holds a Career Award at the Scientific Interface from the Burroughs Wellcome Fund. J.Y. acknowledges support from National Institutes of Health (DP2GM146253). Additional support was provided to R.O. by Wesleyan University Grants in Support of Scholarship funds. W.L.N. acknowledges support from National Institutes of Health (R01AI121337). We thank Drs. Herbert Waite and Günter Wagner for the helpful discussions. We thank Dr. Stavroula Hatzios for providing the Caco-2 cells. We thank Drs. Weimin Zhong, Christopher Waters, and Carey Nadell for suggestions on the manuscript.

## Author contributions

R.O. and J.Y. conceptualized the project. X.H., T.N., K.H.M., Z.J., and J.Y. performed strain construction and validation. X.H., T.N., K.H.M., Z.J., and J.N. performed biofilm imaging and characterization. R.W., A.H., I.G., N.G., and H.Y. performed protein purification and binding assays. R.F.L. and K.S. cultured enteroid monolayer and performed associated characterizations. L.W. and C.C. performed phylogenetic analysis. S.A.M. contributed to the human tissue staining. W.L.N. contributed to the colonization assays. X.H., T.N., S.A.M., W.L.N., K.S., R.O., and J.Y. wrote the manuscript. All authors contributed to the final manuscript.

## Competing interests

Some content of the manuscript (regarding the 57aa) has been included in a provisional patent application, US Application Number: 63/376,414 (pending). Name of Inventors: J.Y. and R.O.; The remaining authors declare no competing interests.
