## [Peer review file · Nature Communications]

Vibrio cholerae biofilms use modular adhesins with glycan-targeting and nonspecific surface binding domains for colonizationEditorial Note: This manuscript has been previously reviewed at another journal that is not operating a transparent peer review scheme. This document only contains reviewer comments and rebuttal letters for versions considered at *Nature Communications*.

Reviewer #1 (Remarks to the Author):

In my view, the authors have done an excellent job in responding to the detailed critiques from the three reviewers. Specifically, the authors have performed a number of substantial additional experiments to address the most serious open questions raised in the reviews. The new results will add substantially to the impact of the paper in the field. First, the new observation that the 57aa loop of Bap1 mediates strong adhesion to lipids is an important new insight into the adhesive role of Bap1. Second, the authors' use of an enteroid model to study the adhesion of *V. cholerae* in a realistic gut setting adds substantially to their claim that the adhesins Bap1 and RbmC (mostly the latter) play important roles in infections of mammalian hosts. I am therefore happy to recommend publication of the revised manuscript in Nature Communications.

Reviewer #2 (Remarks to the Author):

The authors have concisely addressed all of my comments. The revised manuscript text and figures are clear, and do not substantially conflict with other available literature. I recommend publication.

One minor remaining comment: The use of the mouse epithelial enteroid monolayer system (Figure 5) is very interesting, but given the absence of a phenotype in the *in vivo* mouse model (by a prior paper and by the authors of this study), I am a bit doubtful of the conclusion that RbmC and Bap1 critically influence the colonization during infections (lines 311-312). But I think this is not the most important part of the manuscript, and I do not want to delay publication.

Reviewer #1 (Remarks to the Author):

In my view, the authors have done an excellent job in responding to the detailed critiques from the three reviewers. Specifically, the authors have performed a number of substantial additional experiments to address the most serious open questions raised in the reviews. The new results will add substantially to the impact of the paper in the field. First, the new observation that the 57aa loop of Bap1 mediates strong adhesion to lipids is an important new insight into the adhesive role of Bap1. Second, the authors' use of an enteroid model to study the adhesion of *V. cholerae* in a realistic gut setting adds substantially to their claim that the adhesins Bap1 and RbmC (mostly the latter) play important roles in infections of mammalian hosts. I am therefore happy to recommend publication of the revised manuscript in Nature Communications.

Response: We thank the reviewer for the positive assessment of current manuscript.

Reviewer #2 (Remarks to the Author):

The authors have concisely addressed all of my comments. The revised manuscript text and figures are clear, and do not substantially conflict with other available literature. I recommend publication.

Response: We thank the reviewer for the positive evaluation of our revised manuscript.

One minor remaining comment: The use of the mouse epithelial enteroid monolayer system (Figure 5) is very interesting, but given the absence of a phenotype in the in vivo mouse model (by a prior paper and by the authors of this study), I am a bit doubtful of the conclusion that RbmC and Bap1 critically influence the colonization during infections (lines 311-312). But I think this is not the most important part of the manuscript, and I do not want to delay publication.

Response: We thank the reviewer for raising this point. We acknowledge (lines 301-303) that there is an absence of a phenotype in our mouse experiments, cite the prior papers (line 302), and provide a statement (line 303-307) about the limitations of these mouse models for studying *V. cholerae* colonization. We also mention (lines 307-308) future ongoing experiments aimed at improving these models and reconciling results obtained from mouse model and the enteroid model. At this point, we assess that the enteroid model at least shows the effect of Bap1 and RbmC in adhesion in a gut-mimicking environment. Therefore, we did not modify our text in response to this minor comment.